**Report**

# G-quadruplex-binding small molecules ameliorate *C9orf72* FTD/ALS pathology *in vitro* and *in vivo*

Roberto Simone[1,†], Rubika Balendra[1,2,†], Thomas G Moens[1,2], Elisavet Preza[3], Katherine M Wilson[1], Amanda Heslegrave[3], Nathan S Woodling[2], Teresa Niccoli[2], Javier Gilbert-Jaramillo[1,‡], Samir Abdelkarim[4], Emma L Clayton[1], Mica Clarke[1], Marie-Therese Konrad[1], Andrew J Nicoll[1,5], Jamie S Mitchell[1], Andrea Calvo[6], Adriano Chio[6], Henry Houlden[3], James M Polke[7], Mohamed A Ismail[8], Chad E Stephens[8], Tam Vo[8], Abdelbasset A Farahat[8], W David Wilson[8], David W Boykin[8], Henrik Zetterberg[3,9,10], Linda Partridge[2,11] (iD), Selina Wray[3], Gary Parkinson[12], Stephen Neidle[12,*] (iD), Rickie Patani[3,**] (iD), Pietro Fratta[4,***] (iD) & Adrian M Isaacs[1,10,****] (iD)

## Abstract

Intronic GGGGCC repeat expansions in *C9orf72* are the most common known cause of frontotemporal dementia (FTD) and amyotrophic lateral sclerosis (ALS), which are characterised by degeneration of cortical and motor neurons, respectively. Repeat expansions have been proposed to cause disease by both the repeat RNA forming foci that sequester RNA-binding proteins and through toxic dipeptide repeat proteins generated by repeat-associated non-ATG translation. GGGGCC repeat RNA folds into a G-quadruplex secondary structure, and we investigated whether targeting this structure is a potential therapeutic strategy. We performed a screen that identified three structurally related small molecules that specifically stabilise GGGGCC repeat G-quadruplex RNA. We investigated their effect in *C9orf72* patient iPSC-derived motor and cortical neurons and show that they significantly reduce RNA foci burden and the levels of dipeptide repeat proteins. Furthermore, they also reduce dipeptide repeat proteins and improve survival *in vivo*, in GGGGCC repeat-expressing *Drosophila*. Therefore, small molecules that target GGGGCC repeat G-quadruplexes can ameliorate the two key pathologies associated with *C9orf72* FTD/ALS. These data provide proof of principle that targeting GGGGCC repeat G-quadruplexes has therapeutic potential.

**Keywords** amyotrophic lateral sclerosis; C9orf72; frontotemporal dementia; G-quadruplex

**Subject Categories** Neuroscience; Pharmacology & Drug Discovery

See also: **MH Schludi & D Edbauer** (January 2018)

## Introduction

Expansions of a GGGGCC repeat within the first intron of the *C9orf72* gene are the most common genetic cause of both amyotrophic lateral sclerosis (ALS) and frontotemporal dementia (FTD), two rapidly progressive and incurable neurodegenerative disorders (Dejesus-Hernandez *et al*, 2011; Renton *et al*, 2011). While the general population carries < 30 GGGGCC ($G_4C_2$) repeats, with

1   Department of Neurodegenerative Disease, UCL Institute of Neurology, London, UK
2   Department of Genetics, Evolution and Environment, Institute of Healthy Ageing, University College London, London, UK
3   Department of Molecular Neuroscience, UCL Institute of Neurology, London, UK
4   MRC Centre for Neuromuscular Disease, UCL Institute of Neurology, London, UK
5   MRC Prion Unit at UCL, Institute of Prion Diseases, London, UK
6   'Rita Levi Montalcini' Department of Neuroscience, ALS Centre, University of Turin, Turin, Italy
7   Neurogenetics Unit, UCL Institute of Neurology, London, UK
8   Department of Chemistry, Georgia State University, Atlanta, GA, USA
9   Clinical Neurochemistry Laboratory, Institute of Neuroscience and Physiology, Department of Psychiatry and Neurochemistry, The Sahlgrenska Academy, University of Gothenburg, Gothenburg, Sweden
10  UK Dementia Research Institute at UCL, UCL Institute of Neurology, London, UK
11  Max Planck Institute for Biology of Ageing, Cologne, Germany
12  UCL School of Pharmacy, London, UK
    *Corresponding author. Tel: +44 207 7535969; E-mail: s.neidle@ucl.ac.uk
    **Corresponding author. Tel: +44 203 796 0000 Ext 10369; E-mail: rickie.patani@ucl.ac.uk
    ***Corresponding author. Tel: +44 203 4484112; E-mail: p.fratta@ucl.ac.uk
    ****Corresponding author. Tel: +44 207 8375470; E-mail: a.isaacs@ucl.ac.uk
    †These authors contributed equally to this work
    ‡Present address: Facultad de Ciencias de la Vida, Escuela Superior Politécnica del Litoral, ESPOL, Guayaquil, Ecuador

approximately 90% of individuals carrying < 8 repeats, large hexanucleotide repeat expansions (HRE), typically between 800 to > 4,000, are causative of ALS and FTD (Beck *et al*, 2013; van Blitterswijk *et al*, 2013). HREs are transcribed and the resulting RNA forms nuclear foci and can also be translated in all reading frames into dipeptide repeat proteins (DPRs) through a non-canonical process termed repeat-associated non-ATG (RAN) translation (Ash *et al*, 2013; Gendron *et al*, 2013; Lagier-Tourenne *et al*, 2013; Mizielinska *et al*, 2013; Mori *et al*, 2013a,b; Zu *et al*, 2013). Both repeat RNA and DPRs have been proposed to drive pathogenesis: foci can sequester RNA-binding proteins (RBPs) and therefore impair their function (Haeusler *et al*, 2016), while DPRs have been proven to be toxic in numerous disease models (Mizielinska *et al*, 2014; Wen *et al*, 2014; Zhang *et al*, 2016).

$G_4C_2$ RNA can fold to form the highly stable non-canonical G-quadruplex (G-Q) conformation (Fratta *et al*, 2012), a four-stranded structure formed by the stacking of planar tetrads of four non-sequential guanosine residues (G-quartets). RNA G-Qs are able to form *in vivo* (Biffi *et al*, 2014), are enriched in RNA 5′ and 3′ UTRs (Huppert *et al*, 2008) and are known to be involved in regulating numerous RNA functions, including splicing, RNA transport and translation (Simone *et al*, 2015). As G-Qs can directly affect translation (Bugaut & Balasubramanian, 2012), and $G_4C_2$ G-Qs have been shown to specifically sequester disease-relevant RBPs (Haeusler *et al*, 2014; Conlon *et al*, 2016), they may play an important role in both RNA foci and DPR toxicity.

Small molecules binding to both DNA and RNA G-Qs have been identified (Di Antonio *et al*, 2012), and due to the different conformation of RNA and DNA G-Q molecules, ligands preferentially targeting RNA G-Qs have also been developed (Biffi *et al*, 2014). Identification of molecules that specifically target *C9orf72* repeat RNA could have therapeutic potential by shielding pathogenic interactions of the *C9orf72* expanded RNA with RBPs, and/or by interfering with RAN translation. We report here a screen identifying molecules with selectivity for the $G_4C_2$ G-Q RNA and show they are able to reduce both RNA foci formation and RAN translation in *C9orf72* iPSC-neuron models and *C9orf72* repeat-expressing flies.

# Results

### Identification of small molecules that preferentially stabilise RNA $G_4C_2$ G-quadruplexes

In order to identify small molecules that preferentially stabilise RNA $G_4C_2$ G-Qs, we adapted a FRET-based G-Q melting assay (Guyen *et al*, 2004; Collie *et al*, 2012), to specifically report $G_4C_2$ G-Q stabilisation. We have previously identified several novel G-Q-binding chemotypes in the chemical library from the anti-parasitic drug discovery programme at Georgia State University based on non-conjugated aromatic diamidines (Ohnmacht *et al*, 2014). Here, we screened 138 small molecules, 104 from this library and 34 previously established G-Q-binding compounds (Schultes *et al*, 2004; Moore *et al*, 2006). We measured their ability to stabilise $(G_4C_2)_4$ oligonucleotides composed of either RNA or DNA folded into G-Qs. 44/138 small molecules increased the melting temperature ($T_m$) of $(G_4C_2)_4$ RNA by greater than 13°C. Twelve of these showed at least 5°C greater stabilisation of $(G_4C_2)_4$ RNA than $(G_4C_2)_4$ DNA (Fig 1A).

Strikingly, three of these molecules (DB1246, DB1247, DB1273, green circles in Fig 1A) had very similar chemical structures, differing by only two atoms (Fig 1B), indicating a genuine structure–function relationship. No other compounds in the compound set have similar features of two linked five-membered rings.

We therefore took these three small molecules forward for further testing. A stabilisation dose response for both sense $(G_4C_2)_4$ (Fig 1C) and antisense (Appendix Fig S1) $(G_2C_4)_4$ RNA and DNA oligonucleotides confirmed the preferential stabilisation of RNA $G_4C_2$ G-Qs by DB1246, DB1247 and DB1273. Circular dichroism (CD) spectroscopy confirmed that the $(G_4C_2)_4$ RNA formed the expected parallel G-Q structure, with a minimum at 236 nm and a maximum at 264 nm. Each of the three small molecules caused the appearance of a characteristic additional induced CD signal in a separate region of the spectrum (350–550 nm; Fig 1D), while the small molecules alone gave no CD signal (Appendix Fig S2), thus confirming direct binding of DB1246, DB1247 and DB1273 to $(G_4C_2)_4$ RNA G-Qs. We derived the $T_m$ from our CD denaturation curves, which confirmed that $(G_4C_2)_4$ RNA G-Qs were stabilised in the presence of each small molecule (Appendix Fig S3). These compounds bind to the RNA $G_4C_2$ repeat G-Q with high affinities, with $K_d$ values in the range ca 200–400 nM (measurements by fluorescence anisotropy, Appendix Fig S4). These data identify, using FRET, CD and fluorescence anisotropy techniques, three structurally related small molecules DB1246, DB1247 and DB1273 that bind and stabilise RNA $G_4C_2$ G-Qs.

### RNA $G_4C_2$ G-quadruplex-binding small molecules reduce RNA foci in patient iPSC-neurons

We next determined whether the small molecules could alleviate the key pathologies associated with *C9orf72* $G_4C_2$ repeat expansion in patient-derived iPSC-neurons. We first characterised three patient iPSC lines (described in Appendix Table S1). We confirmed the presence of $G_4C_2$ repeat expansions by Southern blotting, which were maintained following differentiation into either motor or cortical neurons (Appendix Fig S5). Cortical neurons were prepared using an established differentiation protocol (Shi *et al*, 2012; Sposito *et al*, 2015). Spinal motor neurons were generated using dual-SMAD and GSK3β inhibition followed by caudal and ventral patterning to the pMN domain and finally promoting cell cycle exit using a Notch antagonist (Hall *et al*, 2017), yielding 90% pure motor neuron cultures (Fig EV1). The efficiency of differentiation did not differ between *C9orf72* and control lines (Fig EV1). We also confirmed that RNA foci were specifically observed in patient iPSC-motor and iPSC-cortical neurons (Appendix Fig S6). We next performed a dose response in one patient iPSC-cortical neuron line to investigate the effect of DB1246, DB1247 and DB1273 on RNA foci formation. For each of the small molecules, a 1 μM treatment for 4 days reduced RNA foci burden (Fig EV2). We therefore treated cortical neurons derived from all three independent *C9orf72* repeat expansion iPSC lines with 1 μM of DB1246, DB1247 or DB1273 for 4 days. Each small molecule significantly reduced RNA foci burden by approximately 50% (Fig 2). The same treatment on iPSC-motor neurons derived from the three independent patient lines showed that DB1246 and DB1273 reduced RNA foci burden, again by approximately 50%, while DB1247 did not significantly reduce RNA foci burden. These data show that small molecules that bind RNA $G_4C_2$ G-Qs can reduce RNA foci in both iPSC-motor and iPSC-cortical neurons.

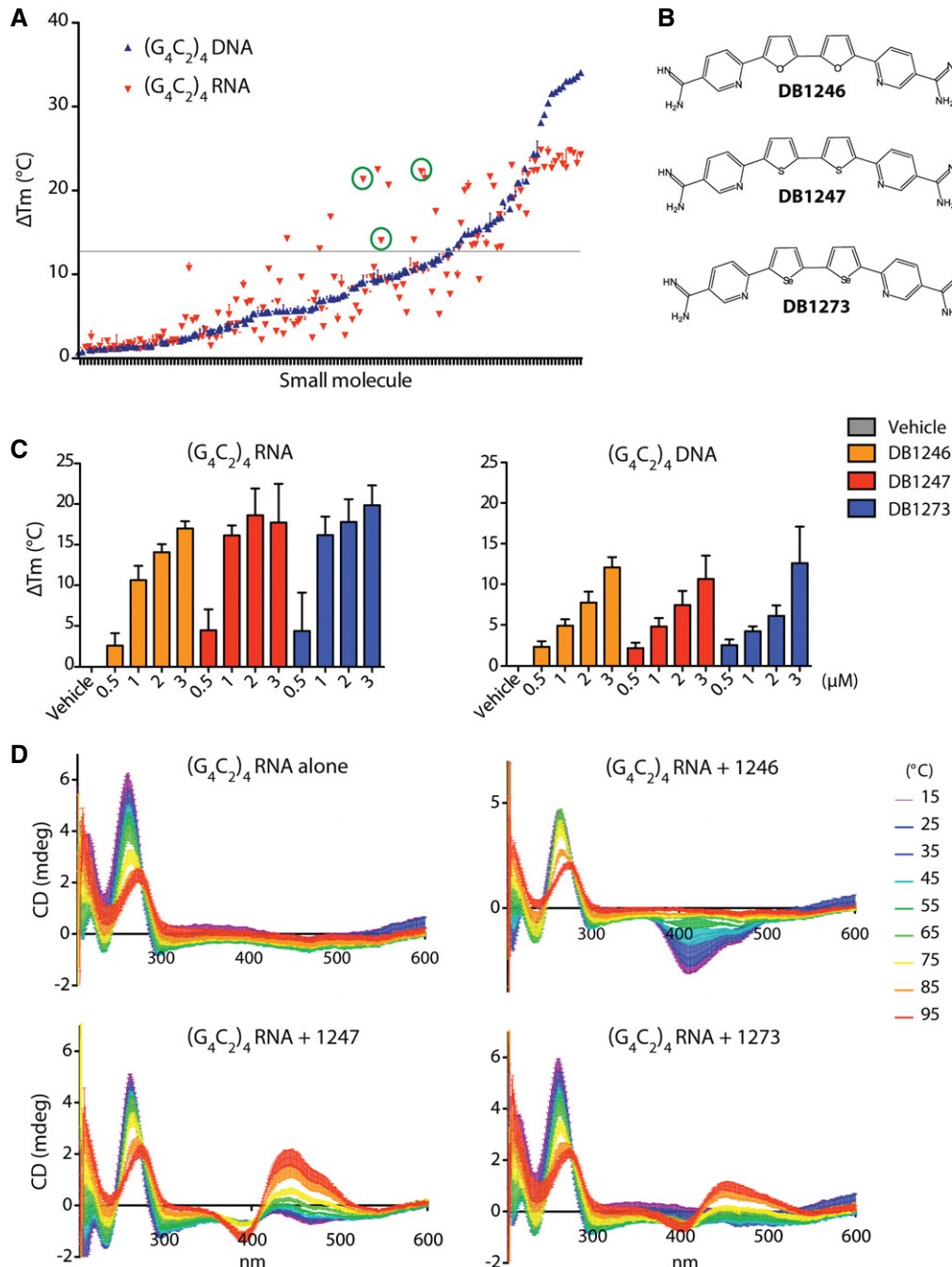

**Figure 1.  Identification and structure of small molecules that preferentially stabilise RNA G₄C₂ repeat G-quadruplexes.**

A   A FRET assay was used to measure the difference in melting temperature ($\Delta T_m$) of $(G_4C_2)_4$ RNA or DNA G-Qs in the presence of 2 μM of 138 different small molecules. An increase in $\Delta T_m$ indicates stabilisation of the G-Q. Small molecules are ranked on the *x*-axis according to their increasing thermal stabilisation of the DNA $(G_4C_2)_4$ G-Q. Small molecules that preferentially stabilise RNA over DNA $(G_4C_2)_4$ G-Qs reside in the upper part of the scatter plot, above the blue curve. An arbitrary $\Delta T_m$ threshold of 13°C greater than vehicle (grey line) and a differential binding to RNA over DNA ($\Delta T_m$RNA–$\Delta T_m$DNA ≥ 5°C) were used to select candidate small molecules.

B   Structures of the three compounds (DB1246, DB1247, DB1273), highlighted by green circles in (A), that show preferential binding to RNA $(G_4C_2)_4$ G-Qs and were further characterised.

C   FRET dose response of DB1246, DB1247 and DB1273 on stabilisation of RNA or DNA $(G_4C_2)_4$ G-Qs.

D   Temperature unfold CD spectra for $(G_4C_2)_4$ RNA alone (which shows a characteristic G-Q structure with minima at 237 nm, maxima at 264 nm and no additional signal), or in the presence of 2 μM DB1246, DB1247 or DB1273. A characteristic induced CD spectrum, in the 350–550 nm region, is observed only in the presence of each small molecule, confirming that each of these three compounds are binding to $(G_4C_2)_4$ RNA G-Qs.

Data information: Data in (A) represent mean ± SD, *n* = 1 with three technical replicates. Data in (C) represent mean ± SD, *n* = 3 independent experiments. Data in (D) represent mean ± SD, *n* = 3 independent experiments.

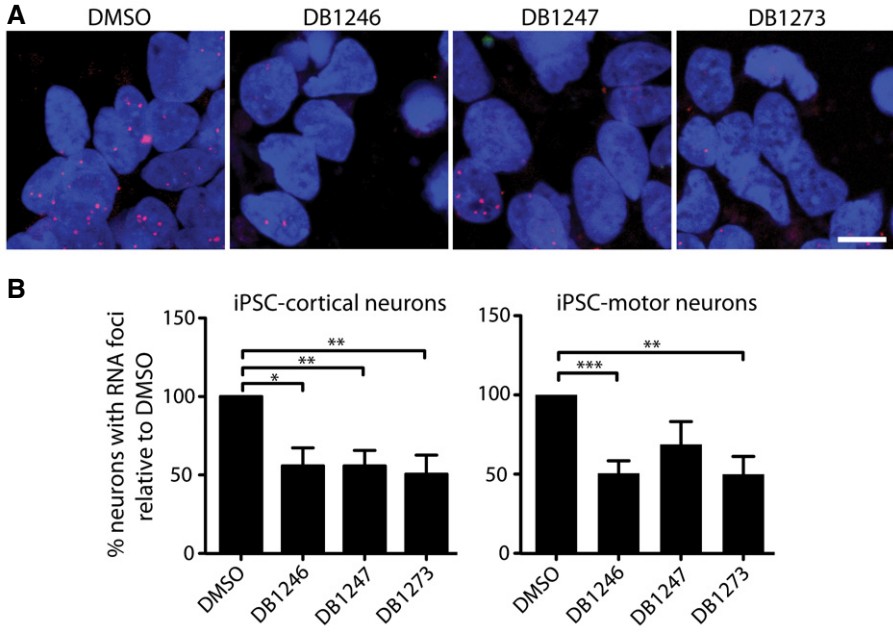

**Figure 2.  G₄C₂ repeat G-quadruplex binding small molecules reduce RNA foci in patient iPSC-cortical and iPSC-motor neurons.**

A   Representative images of *C9orf72* iPSC-cortical neurons treated with DMSO (vehicle control) or 1 μM of DB1246, DB1247 or DB1273, for 4 days. RNA foci are shown in red and nuclei (DAPI) in blue. Scale bar represents 10 μm.

B   Quantification shows RNA foci are significantly reduced by all three compounds in iPSC-cortical neurons and by DB1246 and DB1273 in iPSC-motor neurons. Data are shown as the average percentage of neurons containing RNA foci relative to vehicle (DMSO). $N = 3$ independent *C9orf72* patient lines with two to three inductions per line and at least 70 neurons counted per induction, data are shown as mean and SEM. $*P < 0.05$, $**P < 0.01$, $***P < 0.001$, one-sample two-tailed *t*-test versus normalised control. For cortical neurons, $*P = 0.0124$ (DB1246), $**P = 0.0065$ (DB1247), $**P = 0.0096$ (DB1273). For motor neurons, $***P = 0.0004$ (DB1246), $**P = 0.0030$ (DB1273).

### RNA G₄C₂ G-quadruplex-binding small molecules reduce dipeptide repeat proteins in patient iPSC-neurons without causing toxicity

We next addressed whether DB1246, DB1247 or DB1273 could reduce the other major pathology in C9FTD/ALS—dipeptide repeat proteins. We established an MSD ELISA for poly(GP) and showed that poly(GP) is specifically detected in *C9orf72* repeat expansion iPSC-motor and iPSC-cortical neurons (Appendix Fig S7). Treatment with 1 or 4 μM of DB1246, DB1247 or DB1273 for 4 days did not reduce poly(GP) levels in iPSC-motor neurons (Appendix Fig S8), indicating a differential response of RNA foci and poly(GP) to the small molecules. We therefore investigated 7-day treatments with a range of concentrations (8, 12 and 16 μM). We focused on iPSC-motor neurons due to their shorter differentiation protocol compared to cortical neurons. We found that the two small molecules that reduced RNA foci in iPSC-motor neurons, DB1246 and DB1273, also significantly reduced poly(GP) levels (Fig 3A). DB1273 was the most effective, significantly reducing poly(GP) at all concentrations, with greater than 50% reduction at 16 μM. Importantly, expression levels of *C9orf72* transcripts were not affected by the same treatment (Fig 3B), indicating a direct effect on G₄C₂ repeat G-Q RNA, rather than a more general effect on transcription. We also measured the expression levels of *MCM2* as it has a G-Q within its core promoter region (Huppert & Balasubramanian, 2007), and its expression is reduced by the G-Q-binding small molecule TMPyP₄ (Liu *et al*, 2014). The levels of *MCM2* were unaffected

(Fig 3B), indicating specificity of the small molecules for G₄C₂ RNA G-Qs. We assessed the toxicity of DB1246, DB1247 and DB1273 in iPSC-motor neurons using a concentration range from 0.05 to 40 μM. Importantly, no toxicity was observed in the dose range that reduces poly(GP) levels (Fig 3C). A similar toxicity profile was also observed in human fibroblasts (Appendix Fig S9). These data show that small molecules that bind RNA G₄C₂ G-Qs can specifically reduce the levels of dipeptide repeat proteins generated by endogenous RAN translation of G₄C₂ repeats in cultured patient neurons, without causing toxicity.

### RNA G₄C₂ G-quadruplex-binding small molecules reduce dipeptide repeat proteins and improve survival in GGGGCC repeat-expressing *Drosophila*

We treated *Drosophila* with adult-onset ubiquitous expression of 36 G₄C₂ repeats (Mizielinska *et al*, 2014) for 7 days with the most effective small molecule, DB1273, by feeding it to adult flies in liquid food. While food intake was no different across treatment groups (Appendix Fig S10), DB1273 led to a significant and dose-dependent decrease in poly(GP) levels (Fig EV3A and B). Liquid food did not support the flies long enough to determine whether DB1273 was also able to ameliorate the decreased survival of adult $(G_4C_2)_{36}$ flies. We therefore delivered DB1273 to larvae in solid food, since effects on survival can be observed more rapidly in larvae and larvae eat more than adult flies, allowing delivery of an increased amount of compound. Constitutive, ubiquitous expression of $(G_4C_2)_{36}$ in

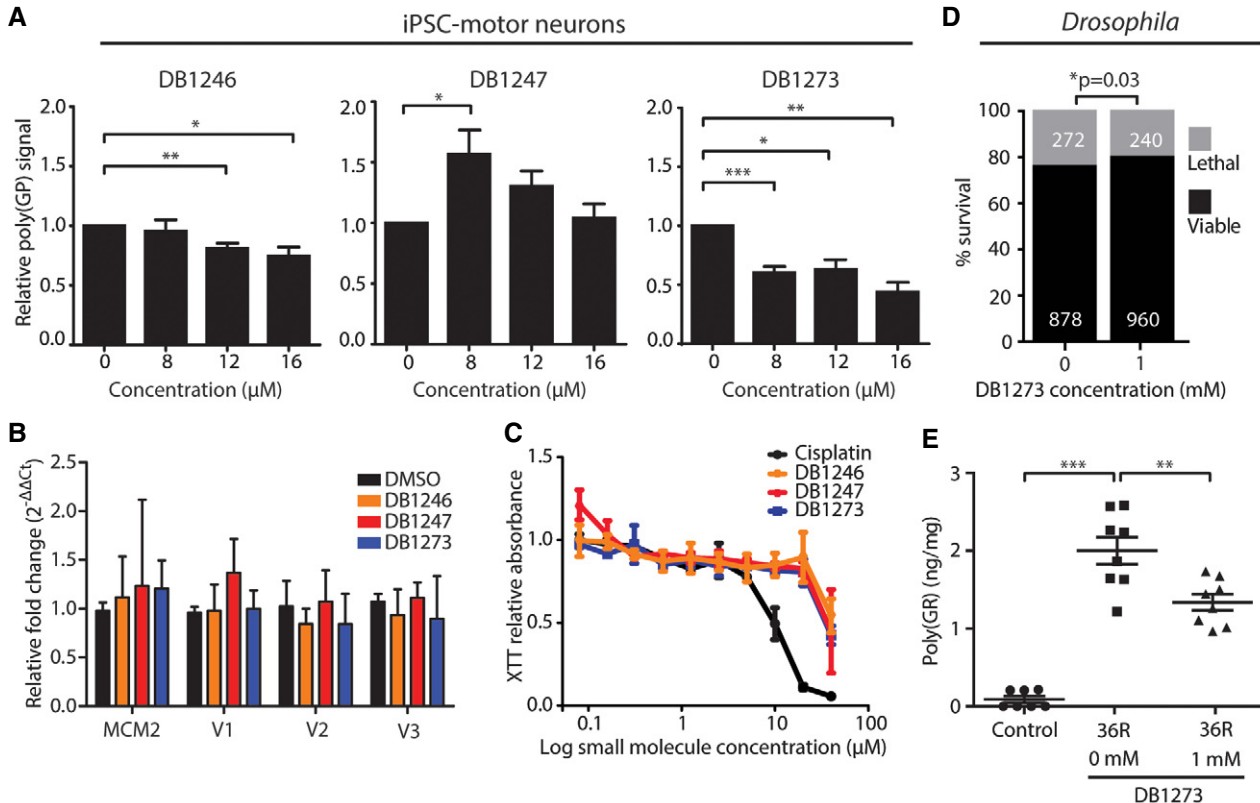

**Figure 3.  G$_4$C$_2$ repeat G-quadruplex binding small molecules reduce poly(GP) in patient iPSC-motor neurons and reduce dipeptide repeat proteins and improve survival in GGGGCC repeat-expressing *Drosophila*.**

A  Poly(GP) levels were measured by MSD immunoassay in *C9orf72* patient iPSC-motor neurons treated for 7 days with 8–16 μM of the G$_4$C$_2$ repeat G-Q-binding small molecules DB1246, DB1247 and DB1273. Treatment with DB1246 or DB1273 leads to a significant reduction in poly(GP) levels relative to vehicle-treated controls. Data are shown as the mean and SEM of three independent *C9orf72* iPSC lines, with one to six differentiations per line. **$P$ = 0.0068 (DB1246, 12 μm), *$P$ = 0.0417 (DB1246, 16 μm), *$P$ = 0.0196 (DB1247, 8 μm), ***$P$ = 0.0002 (DB1273, 8 μm), *$P$ = 0.0194 (DB1273, 12 μm), **$P$ = 0.0062 (DB1273, 16 μm), one-sample two-tailed *t*-test versus normalised control.

B  *C9orf72* patient iPSC-motor neurons were treated for 7 days with 16 μM of each small molecule and the expression levels of *MCM2* and the three *C9orf72* transcript variants (V1–V3) measured by quantitative RT–PCR. Data are shown as the mean and SD of three independent iPSC-motor neuron lines (one induction per line) relative to vehicle (DMSO)-treated controls. No significant changes in gene expression were observed, one-way ANOVA, Dunnett's *post hoc* test, $P$ > 0.05.

C  XTT cell death assay for *C9orf72* iPSC-motor neurons treated for 7 days with 0–40 μM of DB1246, DB1247, DB1273 or cisplatin as a positive control. Data are shown as the mean and SEM of two independent iPSC-motor neuron lines. Toxicity is only observed at the highest dose of 40 μM and not at the penultimate dose of 20 μM.

D  *Drosophila* first-instar larvae expressing 36 G$_4$C$_2$ repeats were placed on food containing either vehicle or 1 mM DB1273, and the number reaching the pupal stage of development counted after 7 days. DB1273 treatment (n = 1,200 flies) significantly improves survival compared to vehicle (n = 1,150 flies), *$P$ = 0.0320, chi-squared test. Data are shown as proportion reaching the pupal stage, with numbers within each group indicated on the bars. Genotype was *w1118; UAS-36(GGGGCC)/+; daGAL4/+ (daGAL4>36R)*.

E  *Drosophila* first-instar larvae ubiquitously expressing 36 G$_4$C$_2$ repeats (36R) were treated with vehicle or 1 mM DB1273 for 5 days, until the third-instar stage (L3), and poly(GR) measured by MSD immunoassay. Poly(GR) was also measured in control (*w1118*) larvae that do not express G$_4$C$_2$ repeats. DB1273 treatment significantly reduced poly(GR) expression, data are shown as mean ± SEM. n = 8 for 36R groups, n = 7 for control group. ***$P$ = 0.0001 (*w1118* versus 36R 0 mM larvae), **$P$ = 0.0019 (0 mM versus 1 mM 36R larvae), one-way ANOVA with Dunnett's *post hoc* test. Genotypes were *w1118*; and *w1118; UAS-36(GGGGCC)/+; daGAL4/+ (daGAL4>36R)*.

*Drosophila* larvae is toxic, with 24% of larvae dying before the pupal stage (Fig 3D). Treatment of first-instar larvae with DB1273 for 7 days led to a significant increase in survival to the pupal stage (Fig 3D). We have shown that poly(GR) is the DPR responsible for toxicity in our G$_4$C$_2$ repeat expansion flies (Mizielinska *et al*, 2014). We therefore developed a poly(GR) MSD immunoassay (Fig EV4). Poly(GR) was specifically detected in (G$_4$C$_2$)$_{36}$ larvae and treatment with DB1273 significantly decreased poly(GR) levels by 33% (Fig 3E). The level of expression of the G$_4$C$_2$ transgene in treated

larvae was not affected (Appendix Fig S11) indicating a specific effect on RAN translation rather than an effect on the transgene itself. In addition, the same treatment in control larvae did not cause toxicity (Appendix Fig S12). We used the inherent fluorescence of DB1273 to examine the biodistribution of DB1273 in our treated larvae. DB1273 was broadly distributed throughout the gut, including the epithelium, while in other tissues, including the central nervous system, infrequent fluorescent puncta were consistently observed in all three replicates (Fig EV5). Overall, these data show

that small molecules that bind RNA $G_4C_2$ G-Qs can reduce dipeptide repeats levels *in vivo* and importantly this leads to functional benefit.

## Discussion

We identify and characterise three small molecules that share structural similarity and the ability to bind and stabilise $G_4C_2$ RNA G-Qs. We demonstrate that these molecules can reduce the frequency of nuclear RNA foci, and the levels of DPRs in *C9orf72* patient iPSC-derived neurons. Furthermore, we provide evidence of *in vivo* efficacy by showing that they can reduce DPRs and improve the survival of GGGGCC repeat-expressing *Drosophila*. Importantly, we observed a significant reduction in poly(GR), the DPR responsible for toxicity in our fly model (Mizielinska *et al*, 2014) as well as other models (Moens *et al*, 2017), and thus likely more clinically relevant than poly(GP). We note that despite limited biodistribution, the small molecule DB1273 was still able to exert a small but significant beneficial *in vivo* effect on survival. This provides proof of principle for targeting G-Q RNA in C9FTD/ALS and indicates that optimising CNS penetrance will lead to even greater efficacy.

DNA and RNA *C9orf72* HREs have been shown to form both G-Qs and hairpin structures (Fratta *et al*, 2012; Reddy *et al*, 2013; Haeusler *et al*, 2014; Su *et al*, 2014; Sket *et al*, 2015). It is widely accepted that HREs exert a toxic gain of function, either linked to RNA-mediated sequestration of RBPs in RNA foci, or through translation into toxic DPRs (Mizielinska & Isaacs, 2014). Previous work has shown that targeting the hairpin conformation of HREs, through small molecules, is a successful approach to reduce both RNA foci and DPRs (Su *et al*, 2014). Here, we demonstrate significant reductions of both RNA foci and DPRs are also possible by targeting the HRE G-Q conformation.

The effect of our small molecules on $G_4C_2$ RNA in patient neurons is consistent with the presence of RNA G-Q structures *in vivo*, as shown using G-Q-specific antibodies (Biffi *et al*, 2014), including specifically in *C9orf72* patient lines (Conlon *et al*, 2016). Interestingly, it has been reported that most predicted RNA G-Qs are unfolded in eukaryotic cells (Guo & Bartel, 2016). As the *C9orf72* HRE represents a very favourable sequence for G-Q formation, our small molecules may enhance and stabilise folding of $G_4C_2$ RNA into G-Qs *in vivo*, as has been reported for other RNA G-Q-binding ligands (Biffi *et al*, 2014). Small molecule-mediated stabilisation of $G_4C_2$ RNA G-Qs could decrease RNA foci either through facilitating their degradation or masking the HRE RNA from the FISH probe, with either scenario indicating effective target engagement. The decrease we observe in DPRs is consistent with reports that G-Qs in the 5′ UTRs of several genes inhibit translation, with inhibition enhanced by the addition of G-Q-binding small molecules (Bugaut & Balasubramanian, 2012; Biffi *et al*, 2014). While our compounds were able to reduce both RNA foci and poly(GP) DPRs, the effect on foci was more rapid and obtained at lower small-molecule concentrations. This may be due to the high stability of poly(GP), which requires 10 days of anti-*C9orf72* antisense oligonucleotide treatment to be significantly reduced in *C9orf72* iPSC-neurons (Gendron *et al*, 2017).

RNA and DNA G-Qs have numerous differences in terms of both conformation, binding partners and regulation, and small molecules have been developed to differentiate between them (Di Antonio

*et al*, 2012). We successfully adopted a DNA/RNA parallel screening approach to identify small molecules preferentially targeting HRE RNA. In support of preferential binding to RNA G-Qs, we observed no impact on transcription levels of *MCM2*, a transcript with a DNA G-Q in its core promoter, which was previously shown to be reduced by $TMPyP_4$ (Liu *et al*, 2014), a generic G-Q-binding molecule that can also bind $G_4C_2$ RNA *in vitro* (Zamiri *et al*, 2014). In summary, these results provide proof of principle for the further development of drugs that selectively bind $G_4C_2$ RNA G-Qs as a therapeutic strategy for C9FTD/ALS.

## Materials and Methods

### FRET G-quadruplex melting assay

The FRET assay was performed as previously described (Guyen *et al*, 2004; Collie *et al*, 2012). Briefly, DNA and RNA dual-labelled oligonucleotides (5′-FAM and 3′-TAMRA) of sequence $(G_4C_2)_4$ and $(G_2C_4)_4$ (IDT, Leuven, Belgium) were initially dissolved as a 100 μM stock in water and diluted to 1 μM in 10 mM sodium cacodylate buffer (pH 7.35) and annealed by heating to 98°C for 10 min, followed by cooling to room temperature in the heating block. Small molecules were stored as 10 mM stock solutions in DMSO; final solutions (2× concentrations) were prepared by dilution in sodium cacodylate buffer (pH 7.35). 96-well plates (Bio-Rad) were prepared by aliquoting 50 μl of the annealed oligonucleotide into each well, followed by 50 μl of the compound solutions, which were tested at both 1 and 2 μM. Fluorescence measurements were made on a DNA Engine Opticon (MJ Research) with excitation at 450–495 nm and detection at 515–545 nm. Fluorescence readings were taken at intervals of 0.5°C over the range 30–100°C, with a constant temperature being maintained for 30 s prior to each reading to ensure a stable value. Final analysis of the data was carried out using a custom script written in Origin 7.0 (OriginLab Corp., Northampton, MA).

### Circular dichroism spectroscopy

Circular dichroism experiments were performed at temperatures between 15°C and 95°C, with a 1°C/min temperature gradient, using a Jasco J715 spectropolarimeter (Jasco Hachioji, Tokyo, Japan) equipped with a Jasco peltier temperature control system. The CD spectrum from 180 to 600 nm was measured for 1 μM pre-folded $(G_2C_4)_4$ RNA, either alone or in the presence of each small molecule (2 μM) in 10 mM sodium cacodylate (pH 7.35). The $T_m$ was calculated by fitting the curve to the Van't Hoff equation using Grafit 5 (Erithacus Software). A CD spectrum of the buffer was recorded and subtracted from all raw signals before plotting using GraphPad Prism 5.

### iPSC-cortical and iPSC-motor neuron cultures

Primary fibroblast lines were generated from skin biopsies, which were obtained under informed consent. Ethical permission for this study was obtained from the National Hospital for Neurology and Neurosurgery and the Institute of Neurology joint research ethics committee. Fibroblasts were reprogrammed as previously described

(Sposito et al, 2015). iPSCs were cultured in feeder-free conditions on Geltrex-coated plates in Essential 8 medium (Thermo Scientific). iPSCs were passaged with 0.5 mM EDTA (Thermo Scientific). Cortical neuron differentiation was as previously described (Shi et al, 2012; Sposito et al, 2015). Briefly, iPSCs were plated at 100% confluency and the media switched to neural induction media (1:1 mixture of N-2 and B-27-containing media supplemented with the SMAD inhibitors Dorsomorphin and SB431452 (Tocris)). Medium consists of DMEM/F-12 GlutaMAX, 1× N-2, insulin, L-glutamine, non-essential amino acids, β-mercaptoethanol, penicillin, streptomycin, Neurobasal, 1× B-27 (Thermo Scientific). Media were changed daily during neural induction, and at the end of the induction period (day 10–12), the converted neuroepithelium was replated onto laminin-coated plates using dispase (Thermo Scientific) and maintained in neural maintenance media (a 1:1 mix of N-2 and B-27) which was replaced every 2–3 days. At around days 25–35, neuronal precursors were passaged further with accutase (Thermo Scientific) and plated for the final time at day 35 onto poly-ornithine and laminin-coated plates (Sigma), before being used in experiments between days 55 and 65. Motor neuron differentiation was as recently described (Hall et al, 2017). Briefly, iPSCs were plated at 100% confluency and the media switched to motor neuron induction media (1:1 mixture of N-2 and B-27-containing media supplemented with the SMAD inhibitors Dorsomorphin and SB431452 (Tocris) and GSK-3β inhibitor CHIR99021). After a 6-day induction period, the converted neuroepithelium was replated onto laminin-coated plates using dispase (Thermo Scientific) and maintained in a 1:1 mix of the described N-2 and B-27 media supplemented with 0.5 μM retinoic acid (Sigma) and 1 μM Purmorphamine (Calbiochem) for a further 7 days, then maintained in a 1:1 mix of the described N-2 and B-27 media supplemented with 0.1 μM Purmorphamine (Calbiochem) for 4 days. Motor neuron precursors were replated onto laminin-coated plates using EDTA (Thermo Scientific) and maintained in a 1:1 mix of the described N-2 and B-27 media supplemented with 0.1 μM compound E (Millipore) before being used at day 30.

### RNA in situ hybridisation

iPSC-derived neurons were fixed in 4% methanol-free formaldehyde (Pierce) for 10 min at room temperature, dehydrated in a graded series of alcohols, air-dried and rehydrated in phosphate-buffered saline (PBS), permeabilised for 10 min in 0.1% Triton X-100 in PBS, briefly washed in PBS and incubated for 30 min in pre-hybridisation solution (40% formamide, 2× SSC, 1 mg/ml tRNA, 1 mg/ml salmon sperm DNA, 0.2% BSA, 10% dextran sulphate, 2 mM ribonucleoside vanadyl complex) at 67°C. Hybridisation solution (40% formamide, 2× SSC, 1 mg/ml tRNA, 1 mg/ml salmon sperm DNA, 0.2% BSA, 10% dextran sulphate, 2 mM ribonucleoside vanadyl complex, 0.2 ng/μl $(C_4G_2)_4$ LNA probe, 5′ TYE563-labelled, Exiqon) was incubated with the cells for 2 h at 67°C. Cells were washed for 30 min at 67°C with high-stringency buffer (2× SSC, 0.1% Triton X-100) and then for 20 min each, in 0.2× SSC buffer. Nuclei were stained by DAPI. Coverslips were then dehydrated in 70% and 100% EtOH and mounted onto slides in Vectashield for iPSC-cortical neurons or Dako mounting medium for iPSC-motor neurons. Images were acquired using an LSM710 confocal microscope (Zeiss) using a plan-apochromat 40×/1.4 NA oil immersion objective.

### Nuclear RNA foci quantification

Five to ten z-stacks were acquired for each field and at least four independent fields were imaged. For iPSC-cortical neurons, maximum intensity projections of each z-stack were analysed in Fiji-ImageJ and nuclear RNA foci automatically counted by using the analyse particles function to identify nuclei and the find maxima function to identify RNA foci in each nucleus. For iPSC-motor neurons, blinded manual counting of maximum intensity projections was performed. At least 70 neurons were counted for each independent differentiation.

### MSD immunoassays

A poly(GP) Meso Scale Discovery (MSD) immunoassay was established using our previously generated rabbit anti-poly(GP) antibody (Mizielinska et al, 2014). A poly(GR) MSD immunoassay was established using newly generated affinity purified rabbit anti-$(GR)_7$ antibodies (Eurogentec). The assays were performed as previously described (Su et al, 2014). Briefly, iPSC-neurons were lysed in RIPA buffer with protease inhibitors (Roche complete mini EDTA-free) and then sonicated using a Soniprep 150 (Renaissance Scientific Limited) probe sonicator. Lysates were centrifuged at 16,000 g to remove insoluble material. Drosophila L3 larvae were frozen in liquid nitrogen, and two larvae per replicate were homogenised in ice-cold RIPA buffer (Sigma) with protease inhibitors (Roche complete mini EDTA-free). Lysis was allowed to proceed on ice for 10 minutes, before lysates were centrifuged at 21,000 g for 20 min at 4°C and supernatant was collected in fresh tubes. The protein concentration of the lysates was determined using the DC protein assay (Bio-Rad) following manufacturer's instructions and equal amounts of protein used in the immunoassay. Capture was performed with either unlabelled anti-poly(GP) or anti-poly(GR) antibodies. Detection on an MSD sector imager utilised either sulfo-tagged anti-poly(GP) antibody or biotinylated anti-poly(GR) antibody followed by sulfo-tagged streptavidin. For iPSC-neurons, prior to analysis, the average reading from a calibrator containing no peptide was subtracted from each reading. For Drosophila larvae, an individual wild-type (w1118) value was chosen at random and subtracted from all samples to correct for background, and resultant negative values were considered to be 0. A four-parameter logistic regression curve was fit to the values obtained using the peptide calibrators using GraphPad Prism, and concentrations interpolated. Specificity was confirmed with a peptide cross-reactivity assay using $(GP)_7$, $(PR)_7$ or $(GR)_7$ synthetic peptides (Biogenes) at a concentration of 100 ng/ml. The lower limit of detection was calculated after fitting a four-parameter logistic regression curve using the MSD workbench 4.0 software.

### XTT assay

iPSC-motor neurons in 96-well plates were treated for 7 days with 0, 0.1, 0.2, 0.3, 0.6, 1.3, 2.5, 5, 10, 20 and 40 μM of DB1246, DB1247, DB1273 or cisplatin as a positive control. Cell viability was measured using the XTT assay kit II (Roche) according to the manufacturer's instructions and absorbance measured with a Tecan Spark 10M plate reader.

## Quantitative RT–PCR

iPSC-derived motor neurons were treated with each of the three small molecules (DB1246, DB1247, DB1273) at 16 µM final concentration for 7 days and compared to the DMSO vehicle. Total RNA was extracted in duplicate for each condition using TRIzol (Invitrogen) and treated with DNase I (Roche). cDNA was synthesised using 1 µg of total RNA for all samples, with a SuperScript III first-strand cDNA synthesis kit (Invitrogen) and an equimolar mixture of oligo dT and random hexamer primers. Real-time qRT–PCR was carried out using Power SYBR Green Master Mix (Applied Biosystems) for *MCM2*. TATA-binding protein (*TBP*) and Glyceraldehyde 3-phosphate dehydrogenase (*GAPDH*) were used as housekeeping genes to normalise across different samples. Additionally, expression of three alternative *C9orf72* transcript variants (V1, V2, V3) was measured using two custom-designed fluorescent LNA PrimeTime® probes (IDT) and previously described isoform-specific primers (Fratta *et al*, 2013). The amplified transcripts were quantified using the comparative Ct method and presented as normalised fold expression ($\Delta\Delta C_t$). Oligonucleotide sequences are provided in Appendix Table S2.

### *Drosophila* stocks and husbandry

All fly stocks were maintained at 25°C on a 12-h:12-h light:dark cycle at 60% constant humidity and on standard sugar–yeast–agar (SYA) medium (agar, 15 g/l; sugar, 50 g/l; autolyzed yeast, 100 g/l; nipagin, 100 g/l; and propionic acid, 2 ml/l). A fly line expressing 36 GGGGCC repeats (36R), under the upstream activating sequence (UAS) promoter, was used (Mizielinska *et al*, 2014). Expression in adults was driven using the inducible daughterless-GeneSwitch (da-GS) driver which was kindly provided by Veronique Monnier (Tricoire *et al*, 2009). Expression in larvae was driven with the constitutive daughterless-GAL4 (daGAL4) driver (Bloomington Drosophila Stock Center).

### Adult *Drosophila* small-molecule treatments

Two days after eclosion, adult *daGS > 36R* flies were fed with liquid food administered via capillaries containing 100 µM RU486 to induce expression of the repeats, or with no RU486 as a control, for 7 days. In addition, flies induced with RU486 were treated with vehicle or DB1273 at 200 µM or 500 µM concentrations. Flies were housed at a density of five flies per vial in plastic vials filled with 2 ml of 1% agar, to ensure humid conditions. Vials were sealed with Parafilm perforated with four holes using a 26-G needle to ensure adequate air circulation. Two graduated 10-µl disposable glass capillary tubes were held in place through the Parafilm using truncated 200-µl pipette tips. The liquid food consisted of 5% (wt/vol) sucrose, 2% yeast extract and blue food dye to aid visualisation of the liquid in the capillaries. A mineral oil overlay was used at the top of the capillary to minimise evaporation. An identical chamber without flies was included to determine evaporative losses. Capillaries were replaced with fresh liquid food as required, and the volume of food ingestion was measured over the course of 7 days. After 7 days, flies were snap-frozen in liquid nitrogen for analysis of abdominal poly(GP) levels by immunoblotting.

**The paper explained**

**Problem**

Amyotrophic lateral sclerosis and frontotemporal dementia are devastating degenerative diseases with no treatments currently available. A mutation in the *C9orf72* gene is the most common cause of both amyotrophic lateral sclerosis and frontotemporal dementia, accounting for approximately 10% of all cases. The *C9orf72* mutation is termed a repeat expansion as it consists of six DNA bases, GGGGCC, that are repeated thousands of times. The GGGGCC repeat DNA is transcribed into repeat RNA, which we have previously shown folds into a distinct secondary structure termed a G-quadruplex. The repeat RNA is then translated into repetitive proteins, termed dipeptide repeat proteins, which are extremely neurotoxic. Therefore, reducing the levels of these dipeptide repeat proteins is a potential therapeutic strategy.

**Results**

We performed a screen to identify compounds that specifically target the *C9orf72* RNA G-quadruplex. We identified three compounds with similar chemical structures that were able to selectively bind the *C9orf72* repeat G-quadruplex RNA. We then treated *C9orf72* patient-induced pluripotent stem cell (iPSC)-derived neurons, and fruit flies harbouring the *C9orf72* repeat expansion, with these compounds. We found that the compounds reduced the levels of the damaging dipeptide repeat proteins in both the iPSC-neurons and the fruit flies. Furthermore, they improved the survival of the *C9orf72* repeat fruit flies.

**Impact**

These results provide proof of principle that targeting the *C9orf72* GGGGCC repeat RNA G-quadruplex is a potential treatment strategy for amyotrophic lateral sclerosis and frontotemporal dementia caused by *C9orf72* repeat expansion.

### Larval small-molecule treatments

To investigate DB1273 toxicity *in vivo*, control daughterless-GAL4 (daGAL4)/+ first-instar larvae were placed into vials containing 90% SYA food, supplemented with vehicle (water and DMSO at a final concentration of 0.3%) or DB1273 to a 0.5 or 1 mM concentration, at a density of 50 larvae per vial, and the number reaching the pupal and adult stage of development were counted. To investigate DB1273 efficacy, *daGAL4 > 36R* first-instar larvae were placed into vials containing 90% SYA food, supplemented with either vehicle or DB1273 to a 1 mM concentration, at a density of 50 larvae per vial. The number reaching the pupal stage of development was counted 7 days later by an observer blinded to the experimental conditions. Third-instar larvae treated with vehicle or DB1273 to a 1 mM concentration were frozen for RNA and protein analysis.

### Immunoblotting

Fly abdomens were homogenised in 20 µl per abdomen of 2× Laemmli sample buffer (4% SDS, 20% glycerol, 120 mM Tris–HCl pH 6.8, 200 mM DTT with bromophenol blue) and heated to 95 °C for 10 min. Samples were centrifuged at 21,000 *g* for 2 min, and supernatants were separated on Nu-PAGE 4-12% Bis–Tris gels in MES running buffer, and then transferred onto nitrocellulose membranes in a Tris–Glycine transfer buffer with 20% methanol. Membranes were incubated with rabbit anti-poly(GP) (Mizielinska *et al*, 2014) 1:1,000 or mouse anti-α-tubulin, Sigma T6199, 1:2,000

followed by horseradish peroxidase-tagged secondary antibody (anti-rabbit HRP, ab6721 or anti-mouse HRP, ab6789, Abcam, 1:10,000). Specific binding was detected with Luminata Forte (Millipore) and imaged with an ImageQuant LAS4000 (GE Healthcare Life Sciences). Intensity of bands was quantified using Fiji-ImageJ Software.

## Statistics

For iPSC-neuron small-molecule effects on RNA foci and poly(GP) levels, we normalised to the control value for each replicate (independent induction) and performed a one-sample $t$-test of treatment replicates versus the control value of 1. For comparison of motor neuron differentiation efficiency between *C9orf72* and control iPSCs, a Mann–Whitney $U$-test was performed due to unequal variances. For *Drosophila* small-molecule treatments, poly(GP) levels were normalised to the loading control, α-tubulin, and a one-way ANOVA performed with Dunnett's *post hoc* test across the treatment groups. For comparing lethality/viability in *Drosophila*, a chi-square test was used. For quantitative PCR, a $t$-test was used for comparison of two groups and a one-way ANOVA with Dunnett's *post hoc* test for comparison of more than two groups. Statistical analyses were performed in GraphPad Prism 5.

**Expanded View** for this article is available online.

## Acknowledgements

We thank Professor Chris Shaw for providing *C9orf72* iPSC line 3 and Lucy Minkley for technical assistance. This work was funded by the Thierry Latran Foundation (AMI, PF, GP), ERC (AMI, H2020-ERC-2014-CoG-648716), MRC (PF, MR/M008606/1), Brain Research Trust (TGM), Alzheimer's Research UK (SW, AMI, ARUK-PPG2012B-13), Leonard Wolfson Foundation (AJH), NIH (GM111749 to W. D. W. and D. W. B), and Clinical Research in ALS and Related Disorders for Therapeutic Development (CReATe), which is funded through a collaboration between NCATS and National Institute of Neurological Disorders and Stroke. RB is a Leonard Wolfson Clinical Research Training Fellow and funded by a Wellcome Trust Research Training Fellowship (107196/Z/14/Z). HZ is a Wallenberg Academy Fellow. SW and EP are supported by the NIHR Queen Square Dementia Biomedical Research Unit. RP is a Wellcome Trust Intermediate Clinical Fellow (101149/Z/13/A). Some of this work was undertaken at UCLH/UCL who received a proportion of funding from the Department of Health's NIHR Biomedical Research Centres funding scheme.

## Author contributions

RS, RB, TGM, RP, AH, ELC, NSW, TN, AJN, JMP, WDW, DWB, HZ, LP, SW, GP, SN, RP, PF and AMI designed and/or interpreted aspects of the different experiments. RS, RB, TGM, EP, KMW, NSW, TN, JGJ, SA, MC, M-TK, AJN, JSM, MAI, CES, TV, AAF, ACa, ACh, HH, SW and PF performed experiments. RS, RB, PF and AMI wrote the manuscript with input from all co-authors.

## Conflict of interest

The authors declare that they have no conflict of interest.

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
