## [Review Process File · EMBO Molecular Medicine]

G-quadruplex-binding small molecules ameliorate C9orf72 FTD/ALS pathology in vitro and in vivo

Roberto Simone, Rubika Balendra, Thomas G. Moens, Elisavet Preza, Katherine M. Wilson, Amanda Heslegrave, Nathan S. Woodling, Teresa Niccoli, Javier Gilbert-Jaramillo, Samir Abdelkarim, Emma L. Clayton, Mica Clarke, Marie-Therese Konrad, Andrew J. Nicoll, Jamie Mitchell, Andrea Calvo, Adriano Chio, Henry Houlden, James M. Polke, Mohamed A. Ismail, Chad E. Stephens, Tam Vo, Abdelbasset A. Farahat, W. David Wilson, David W. Boykin, Henrik Zetterberg, Linda Partridge, Selina Wray, Gary Parkinson, Stephen Neidle, Rickie Patani, Pietro Fratta & Adrian M. Isaacs

Corresponding authors:

Stephen Neidle, UCL School of Pharmacy

Rickie Patani, UCL Institute of Neurology

Pietro Fratta, UCL Institute of Neurology

Adrian M. Isaacs, UCL Institute of Neurology

Review timeline:

Submission date:	30 March 2017
Editorial Decision:	11 May 2017
Revision received:	19 August 2017
Editorial Decision:	20 September 2017
Revision received:	05 October 2017
Accepted:	09 October 2017

Editor: Céline Carret

Transaction Report:

1st Editorial Decision

11 May 2017

Thank you for the submission of your manuscript to EMBO Molecular Medicine. We have now heard back from the three referees whom we asked to evaluate your manuscript.

You will see that the three referees find the study interesting and timely, with the main concern being similarly noted by both referees 1 and 3 who suggest performing supplementary analyses in flies to verify the compound effect on the eye phenotype and /or survival.

We would welcome the submission of a revised version for further consideration and depending on the nature of the revisions, this may be sent back to the referees for another round of review.

Please note that EMBO Molecular Medicine encourages a single round of revision and that, as acceptance or rejection of the manuscript will depend on another round of review, your responses should be as complete as possible.

I look forward to receiving your revised manuscript.

***** Reviewer's comments *****

Referee #1 (Remarks):

In this manuscript, the authors identified unique small molecules that stabilize the G-quadruplex structure of C9ORF72 G4C2 repeat expansions. Using iPSC-differentiated neurons and *Drosophila* that express G4C2 repeat expansions, the authors show the identified small molecules reduce both sense RNA foci and DPR proteins, two pathological features produced by G4C2 repeat expansions. The study is very interesting and timely as research teams in both academia and industry are deeply engaged in this area of preclinical space. Overall the manuscript is supported by compelling data with only minor comments noted. The specific points are list below.

1. In Figure 3D and E, the authors show compound 1273 decreases poly(GP) DPR protein in (G4C2)₃₆ *Drosophila* model. Using this model, the authors should examine 1) whether compound 1273 can rescue the eye degeneration and impact lifespan; 2) whether compound 1273 influences RNA foci formation and/or c9 transcript levels.
2. It appears that the identified small molecules also bind to antisense G2C4 repeat expansions (Figure S1). The authors should examine whether these compounds can reduce antisense RNA foci in iPSC-differentiated neurons or cell culture systems.

Referee #2 (Remarks):

Simone et al. report three structurally related compounds that reduce RNA foci and translation into poly-GP in iPSC and fly models. These compounds are the result of an in vitro screen of 138 preselected compounds for binding to the G4C2 RNA repeat and specifically stabilize the G-quartet structure. This exciting work goes all the way from compound screening, hard-core biochemistry to cellular and animals models. The compounds could be a lead for further optimization prior to clinical application in C9orf72 patients and should be interesting for the EMBO Mol Med readership.

The manuscript is well written and includes already all the major controls. Some small experiments could improve the manuscript further:

- The authors speculate that the compounds may either prevent binding of RBPs or facilitating degradation to the RNA. qPCR could be useful to distinguish between masking of the repeat RNA for FISH and actual degradation. How are antisense foci affected by the compounds given the preferential binding to sense RNA?
- Can DB1273 reduce repeat toxicity in the fly model? Does it have toxic side effects in flies at the used concentration?
- Figure S8 should be merged with Figure 2, because it contains key data.

Referee #3 (Remarks):

Simone et al., reported that G-quadruplex-binding small molecules obtained from aromatic diamidines ameliorated RNA foci formation and poly(GP) levels which were provoked by disease-associated G4C2 repeats in C9orf72. Three similar-structured compounds, DB1246, DB1247, and DB1273 were able to decrease the number of RNA foci in human iPSC-derived cortical/motor neurons which harbored mutant C9orf72. Moreover, the authors demonstrated that administration of those molecules reduced the expression levels of poly(GP) in a fly model of C9FTD/ALS. The presented data seems enough to convince the beneficial effects of the compounds at least on

silencing "aggregation formation" of both RNA and DPR(s). However, the manuscript did not contain the data of phenotypic change(s) or the molecular mechanism which weaken the impact of the manuscript. Since the authors have previously shown that the same fly model harboring mutant C9orf72 exhibited neurodegeneration and shorter survival, it would be necessary to investigate whether the compounds have beneficial effects on survival and/or neurodegeneration in the fly model.

Major issues

1. As described above, the authors need to show whether the compounds have beneficial effects on the survival and/or eye degeneration observed in C9FTD/ALS fly model they used. The previous work from the same group has clearly shown the model had a neurodegeneration phenotype (Mizielinska et al., 2014).
2. In the discussion section authors described, "Decreased RAN translation could occur through impeding either assembly of the ribosomal pre-initiation complex (PIC), or subsequent PIC scanning of the RNA". Addressing this issue, the authors can enhance the quality of the manuscript which is required for the journal.
3. It is necessary to investigate whether the compounds can reduce the expression levels of other DPRs including poly(PR), poly(GR), and/or poly(GA).
4. It would be helpful if the authors describe more biological signature of the compounds, such as tissue distribution.

Minor issues

1. In Figure 1B, "DB-1273" should be "DB1273".
2. In P4 L12, the authors described, "three of these molecules had very similar chemical structures, differing by only two atoms, indicating a genuine structure-function relationship." I just wondered if there were no similar chemical structured observed in the other 135 molecules.
3. In Figure S3, there are two 'RNA-DB1247'. The last one seems to be RNA-DB1273.
4. It would be helpful if the authors add the result of RNA-DB1273 to Figure S4.

Mizielinska, S., Gronke, S., Niccoli, T., Ridler, C.E., Clayton, E.L., Devoy, A., Moens, T., Norona, F.E., Woollacott, I.O., Pietrzyk, J., et al. (2014). C9orf72 repeat expansions cause neurodegeneration in *Drosophila* through arginine-rich proteins. *Science* 345, 1192-1194.

1st Revision - authors' response

19 August 2017

We thank the reviewers for their helpful comments, which have considerably strengthened our findings; our point by point response is below. We are particularly excited that the one consistent request across the three reviewers – to investigate the effect of DB1273 on the survival of our (G4C2)₃₆ flies – was performed and shows a significant improvement. We are also very excited that we were able to show that this treatment significantly reduces poly(GR), the toxic DPR in our flies. As far as we are aware, this is the first report of a potential therapeutic strategy for C9FTD/ALS that has shown a significant and quantitative reduction in the toxic DPR poly(GR). As such we feel the manuscript is now substantially improved and hope it is now acceptable for publication in EMBO Molecular Medicine.

Reviewer 1

1. In Figure 3D and E, the authors show compound 1273 decreases poly(GP) DPR protein in (G4C2)₃₆ *Drosophila* model. Using this model, the authors should examine
 - 1) whether compound 1273 can rescue the eye degeneration and impact lifespan;

Response

We treated our (G4C2)₃₆ *Drosophila* with DB1273 and observed a significant increase in survival, indicating it has a beneficial functional effect in vivo, now shown in Fig. 3D. We thank the reviewer for raising this important point as these new findings considerably strengthen the manuscript.

- 2) whether compound 1273 influences RNA foci formation and/or c9 transcript levels.

Response

We agree that this is an important control. We performed quantitative real-time PCR and show that DB1273 does not influence C9orf72 transcript levels in our treated (G4C2)₃₆ Drosophila, now described in the Fig. S11. This indicates the reduction in DPRs is a specific inhibition of RAN translation rather than a decrease in transcription of the transgene and further strengthens our findings.

2. It appears that the identified small molecules also bind to antisense G2C4 repeat expansions (Figure S1). The authors should examine whether these compounds can reduce antisense RNA foci in iPSC-differentiated neurons or cell culture systems.

Response

We agree this is an interesting experiment. Unfortunately our antisense FISH protocol identified only low numbers of antisense foci in iPSC-neurons so we were unable to determine whether the small molecules had an effect. We feel that while interesting, this experiment is not essential for underpinning our main conclusions and can be addressed in future studies.

Reviewer 2

1. The authors speculate that the compounds may either prevent binding of RBPs or facilitating degradation to the RNA. qPCR could be useful to distinguish between masking of the repeat RNA for FISH and actual degradation. How are antisense foci affected by the compounds given the preferential binding to sense RNA?

Response

As stated above, our antisense FISH protocol identified only low numbers of antisense foci in iPSC-neurons so we were unable to determine whether the small molecules had an effect. The qPCR experiment would also be interesting, but due to our focus on the in vivo experiments we were not able to do this in time. We feel that these experiments, while interesting, are not essential for underpinning our main conclusions and can be addressed in future studies.

2. Can DB1273 reduce repeat toxicity in the fly model?

Response

We agree this is a key question. As described above, we now show that DB1273 can reduce toxicity in our fly model, now described in Fig. 3D, these data significantly enhance our findings.

3. Does it have toxic side effects in flies at the used concentration?

Response

We tested DB1273 in control flies at the used concentration and we do not observe toxicity, now described in Fig. S12.

4. Figure S8 should be merged with Figure 2, because it contains key data.

Response

We have now promoted Figure S8 to an expanded view figure (Fig. EV2) to make it more visible.

Reviewer 3

1. As described above, the authors need to show whether the compounds have beneficial effects on the survival and/or eye degeneration observed in C9FTD/ALS fly model they used. The previous work from the same group has clearly shown the model had a neurodegeneration phenotype (Mizielinska et al., 2014).

Response

We agree this is a key question. As described above, we now show that DB1273 can reduce toxicity in our fly model, now described in Fig. 3D, which significantly enhances our findings.

2. In the discussion section authors described, "Decreased RAN translation could occur through impeding either assembly of the ribosomal pre-initiation complex (PIC), or subsequent PIC scanning of the RNA". Addressing this issue, the authors can enhance the quality of the manuscript which is required for the journal.

Response

We agree that this would be very interesting to determine, however, currently the mechanism(s) of RAN translation are not known. Therefore we feel it is beyond the scope of

this work to define this mechanism and how the small molecules specifically inhibit it. We have therefore removed this speculation from the discussion.

3. It is necessary to investigate whether the compounds can reduce the expression levels of other DPRs including poly(PR), poly(GR), and/or poly(GA).

Response

We agree that particularly measuring the levels of poly(GR) would strengthen our findings as poly(GR) is the toxic DPR in our flies and has widely been shown to be toxic. We therefore developed a sensitive and specific poly(GR) MSD immunoassay, now described in Figure EV4. We used this assay to show that poly(GR) is significantly reduced, in vivo, in *Drosophila* treated with DB1273, now described in Fig. 3E.

We note that even though poly(GR) is thought to be one of the toxic DPRs, we are not aware of any other study that has quantitatively shown a reduction in poly(GR) through any therapeutic approach, (largely because of the difficulty in detecting poly(GR)). We therefore feel that our demonstration of a reduction in poly(GR) is a step forward for the field and considerably strengthens our findings.

4. It would be helpful if the authors describe more biological signature of the compounds, such as tissue distribution.

Response

We have now investigated the tissue distribution of DB1273, now described in figure EV5. We show broad distribution in the gut with low but consistent presence in the central nervous system. We also discuss this finding further in the discussion

Minor issues

1. In Figure 1B, "DB-1273" should be "DB1273".

Response

We thank the reviewer for spotting this, we have now corrected it.

2. In P4 L12, the authors described, "three of these molecules had very similar chemical structures, differing by only two atoms, indicating a genuine structure-function relationship." I just wondered if there were no similar chemical structures observed in the other 135 molecules.

Response

We thank the reviewer for raising this point – no other compound had a similar structure, we therefore now state in the results section ‘No other compounds in the compound set have similar features of two linked five-membered rings’.

3. In Figure S3, there are two 'RNA-DB1247'. The last one seems to be RNA-DB1273.

Response

Thanks again we have now corrected.

4. It would be helpful if the authors add the result of RNA-DB1273 to Figure S4.

Response

We were not able to perform fluorescence anisotropy on DB1273 due to its lower inherent fluorescence. We have clarified this by stating in the fluorescence anisotropy methods ‘Fluorescence anisotropy was performed for DB1246 and DB1247 but was not possible for DB1273 due to its weaker fluorescence’.

2nd Editorial Decision

20 September 2017

Thank you for the submission of your revised manuscript to EMBO Molecular Medicine. We have now received the enclosed reports from the referees that were asked to re-assess it. As you will see the reviewers are now fully supportive and I am pleased to inform you that we will be able to accept your manuscript pending the following final amendments:

- I could not find any details about how the iPSC were derived from and any ethical statement regarding patients. Can you please refer to our guidelines and provide the missing information?

Please submit your revised manuscript within two weeks. I look forward to seeing a revised form of your manuscript as soon as possible.

***** Reviewer's comments *****

Referee #1 (Remarks for Author):

The authors have addressed all the previous concerns and is markedly improved. The paper is now suitable for publication.

Referee #2 (Comments on Novelty/Model System for Author):

The authors fully addressed all the critical points and provide now in vivo data for the efficacy of the new compound, which correlates with the new poly-GR ELISA data. The DB1237 compound is an interesting candidate for further development to increase brain delivery in vivo. I fully recommend publication in EMM.

Referee #3 (Remarks for Author):

The authors adequately answered requested questions by performing additional experiments. The manuscript is now suitable for publication in EMBO Molecular Medicine.

Corresponding Author Name: Adiliah Isaacs
Journal Submitted to: EMBO Molecular Medicine
Manuscript Number: EMM-2017-07850